# Semisynthetic fluorescent pH sensors for imaging exocytosis and endocytosis

Magalie Martineau [1,2], Agila Somasundaram[3], Jonathan B. Grimm[4], Todd D. Gruber[4], Daniel Choquet [1,2,5], Justin W. Taraska [3], Luke D. Lavis [4] & David Perrais [1,2]

The GFP-based superecliptic pHluorin (SEP) enables detection of exocytosis and endocytosis, but its performance has not been duplicated in red fluorescent protein scaffolds. Here we describe "semisynthetic" pH-sensitive protein conjugates with organic fluorophores, carbofluorescein, and Virginia Orange that match the properties of SEP. Conjugation to genetically encoded self-labeling tags or antibodies allows visualization of both exocytosis and endocytosis, constituting new bright sensors for these key steps of synaptic transmission.

[1] University of Bordeaux, F-33000 Bordeaux, France. [2] Centre National de la Recherche Scientifique, Interdisciplinary Institute for Neuroscience, UMR 5297, F-33000 Bordeaux, France. [3] National Heart, Lung, and Blood Institute, US National Institutes of Health, Bethesda, MD 20892, USA. [4] Janelia Research Campus, Howard Hughes Medical Institute, 19700 Helix Drive, Ashburn, VA 20147, USA. [5] Bordeaux Imaging Center, UMS 3420 CNRS, Université de Bordeaux, US 4 INSERM, F-33000 Bordeaux, France. Magalie Martineau, Agila Somasundaram, Jonathan B. Grimm, Justin W. Taraska, Luke D. Lavis, and David Perrais contributed equally to this work. Correspondence and requests for materials should be addressed to J.W.T. (email: justin.taraska@nih.gov) or to L.D.L. (email: lavisl@janelia.hhmi.org) or to D.P. (email: david.perrais@u-bordeaux.fr)

Synaptic transmission is mediated by the rapid fusion of synaptic vesicles (SVs) with the plasma membrane. Precise monitoring of exocytosis is important for elucidating fundamental mechanisms of cell–cell communication, and investigating the underlying causes of neurological disorders[1]. The lumen of synaptic vesicles are typically acidified (pH 5.6) by the action of vesicle-resident V-ATPases, which creates the driving force for neurotransmitter uptake. Upon fusion with the plasma membrane, the contents of the vesicle rapidly equilibrate with the extracellular environment (pH 7.4). This large change in pH allows for the visualization of exocytosis using a pH-sensitive variant of green fluorescent protein (GFP) that is expressed as a fusion with a vesicular membrane protein[2]. This "supercliptic pHluorin" (SEP) exhibits ideal properties for detecting the change in pH upon vesicle fusion, with near-ideal $pK_a$, cooperative protonation, and low background fluorescence in the protonated state, making it an excellent tool for monitoring exocytosis in living cells[2].

A useful extension of this technology has been the creation of pH sensors based on red fluorescent proteins (RFPs) such as mOrange2[3], pHTomato[4], pHoran4[5], and pHuji[5]. Longer excitation wavelengths are less phototoxic, elicit lower levels of autofluorescence, facilitate multicolor imaging experiments, and allow concomitant use of optogenetics. Nevertheless, it has proven difficult to engineer red-shifted pHluorins that match the optimal $pK_a$, cooperativity, and dynamic range of SEP, perhaps due to inherent limitations in RFP scaffolds. More generally, these techniques rely on overexpression of reporter proteins in SVs and

the effect of overexpression is a confounding factor in interpreting experimental results[6].

To circumvent the problems with genetically encoded red fluorophores, we develop a "semisynthetic" sensor platform using red-shifted chemical fluorophores derived from fluorescein—carbofluorescein (CFl) and Virginia Orange (VO)—with pH sensitivities similar to SEP. We have adapted these fluorophores to serve as probes for exo- and endocytosis in two ways. First, we synthesize benzylguanine derivatives, which bind specifically to SNAP-tag ligands[7], to label exogenous vesicular proteins fused with the SNAP-tag. Second, we label an antibody targeted to the extracellular epitope of a vesicular protein, synaptotagmin1, which allow the imaging of the exo-/endocytosis cycle of an endogenous protein.

## Results

**In vitro characterization of CFl and VO as pH sensors.** Given the limitations of pH-sensitive RFPs and the potential problems with overexpression of sensor proteins, we pursued an alternative strategy: creation of semisynthetic pH indicators using organic pH-sensitive dyes attached to either expressed self-labeling tags such as the SNAP-tag[7] or antibodies that recognize native vesicular proteins (Fig. 1a). To match the performance of SEP, we required a pH-sensitive organic dye that can undergo a cooperative transition from a bright, fluorescent form at neutral pH to a nonfluorescent form at low pH. Unfortunately, the majority of pH-sensitive dyes do not meet these requirements. The

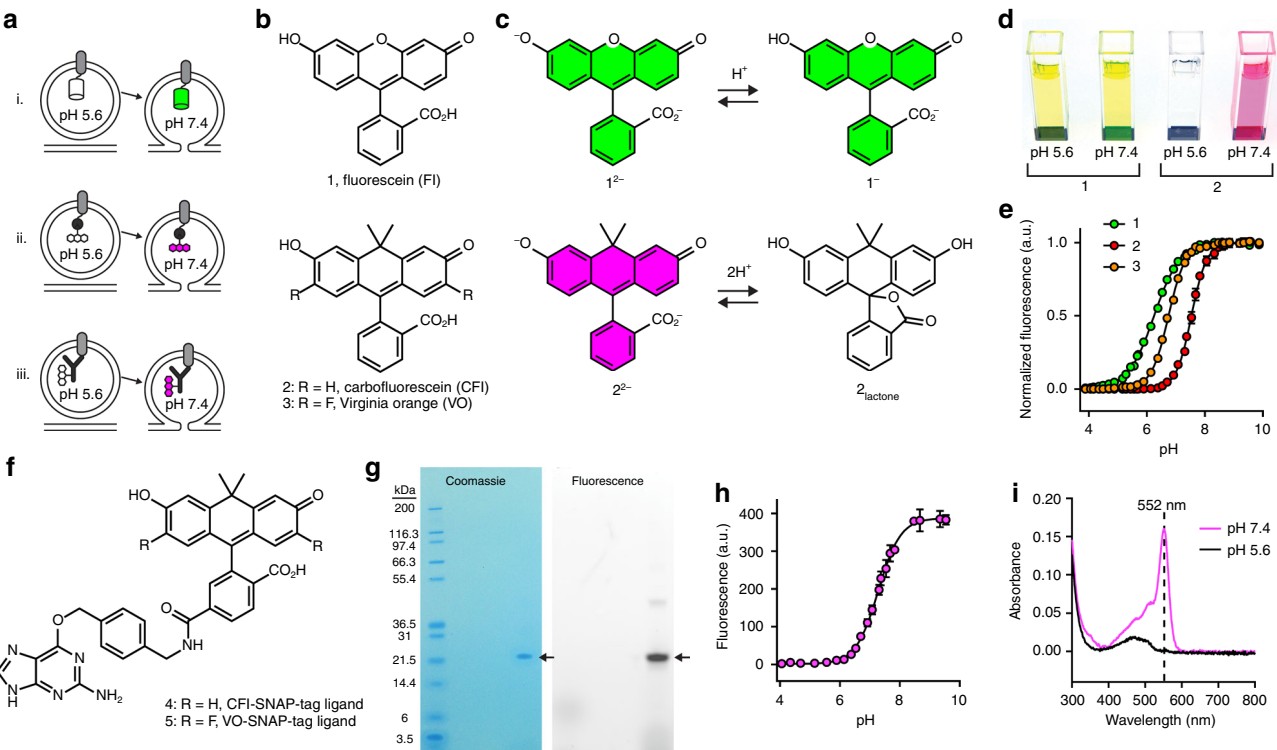

**Fig. 1** Design and characterization of semisynthetic fluorescent reporters for exocytosis. **a** Three protein labeling strategies for imaging exocytosis. Cartoons depicting vesicles expressing pH-sensitive fluorescent protein, such as SEP or pHuji, fused to a SV protein (i) or SNAP-tag enzyme fused to an SV protein and labeled with a pH-sensitive organic fluorophore (ii) and labeling of endogenous SV proteins with an antibody conjugated to a pH-sensitive dye (iii). **b** Chemical structures of Fl (1), CFl (2), and VO (3). **c** Fl and CFl respond differently to pH. Fluorescein undergoes noncooperative protonation from the dianion ($1^{2-}$) to the monoanion ($1^{-}$), whereas carbofluorescein undergoes cooperative protonation from the highly colored dianion ($2^{2-}$) to a colorless lactone form (2lactone). **d** Image of cuvettes containing compounds 1 or 2 (5 µM) at pH 5.6 and pH 7.4. **e** Plot of fluorescence vs. pH for compounds 1, 2, and 3. **f** Chemical structure of CFl–SNAP-tag ligand (4) and VO–SNAP-tag ligand (5). **g** PAGE of SNAP-tag protein labeled with compound 4 and visualized by Coomassie staining or by fluorescence (indicated by arrows). **h** Plot of fluorescence vs. pH for 4–SNAP-tag conjugate. $pK_a = 7.3$ and $\eta_H = 1.2$. **i** Absolute absorbance of 4–SNAP-tag conjugate (~3 µM) at pH 7.4 (magenta) and pH 5.6 (black). Error bars represent s.d. for 4 experiments each

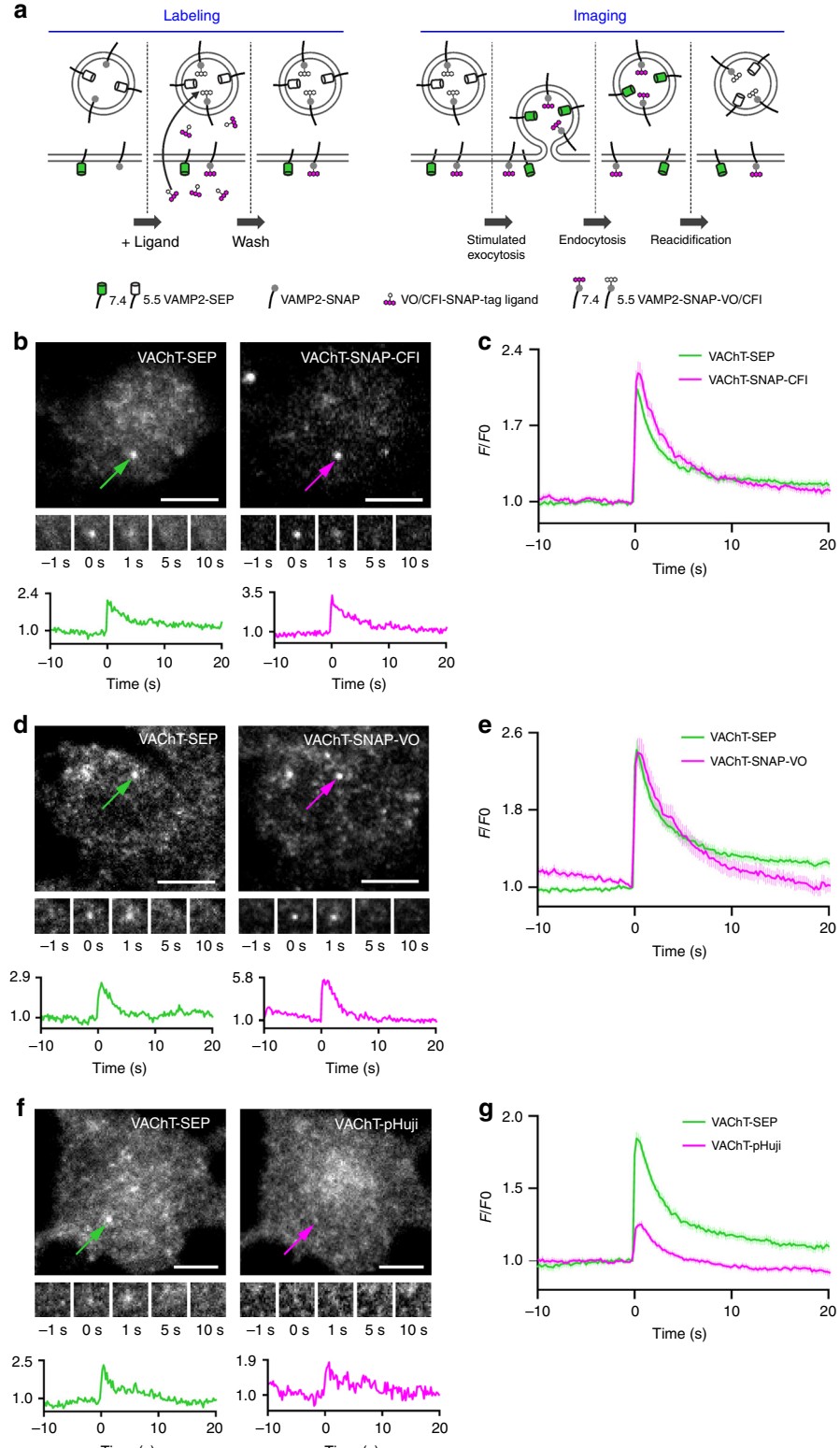

**Fig. 2** Detection of single exocytosis events in PC12 cells. **a** Experimental scheme: prior to imaging, cell-permeable VO/CFI-SNAP-tag ligand is incubated with transfected neuronal cultures to specifically label VAMP2-SNAP. Both VAMP2-SEP and VAMP2-SNAP-VO/CFI are quenched at acidic pH and exhibit maximal fluorescence at the surface. **b**, **d**, **f** TIRF microscopic images of PC12 cells expressing VAChT-SEP (left) and VAChT-red shifted indicator (right) undergoing fusion. Images of whole cell with a fusion event indicated with arrow (top); time-lapse of the marked exocytic event (middle, 3.5 × 3.5 μm); normalized intensity traces for the marked event in the green and red channels (bottom); scale bars: 5 μm. **b** VAChT-SNAP-CFI. **d** VAChT-SNAP-VO. **f** VAChT-pHuji. **c**, **e**, **g** Plots of normalized fluorescence vs. time averaged across single-vesicle fusion events for vesicles labeled with VAChT-SEP (green) or VAChT-red-shifted indicator (magenta); error bars show ± s.e.m. **c** VAChT-SNAP-CFI, $n = 113$ events from 5 cells. **e** VAChT-SNAP-VO, $n = 116$ events from 10 cells. **g** VAChT-pHuji, $n = 126$ events from 3 cells

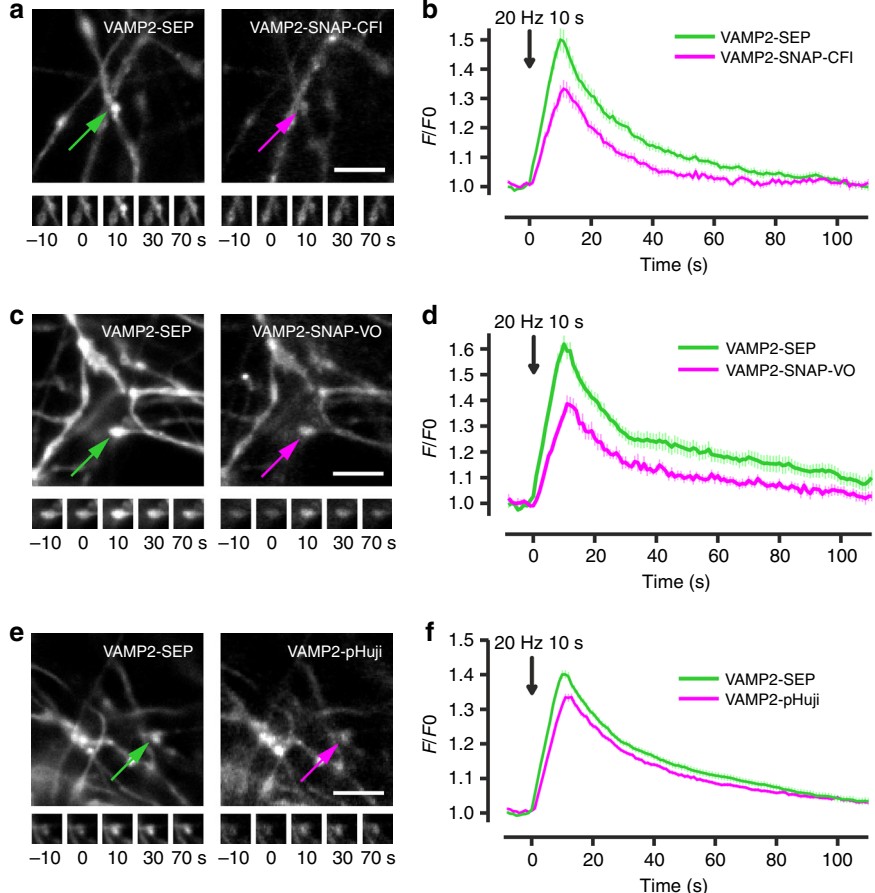

**Fig. 3** Detection of synaptic vesicle exocytosis and recycling in hippocampal neurons. **a** Fluorescence images of hippocampal axons expressing VAMP2-SEP and VAMP2-SNAP-CFl with the locations of vesicle fusion indicated with arrows (top); scale bar: 5 μm. Time-lapse images of the exocytosis events (bottom). **b** Average VAMP2-SEP (green) and VAMP2-SNAP-CFl (magenta) fluorescence signals in response to field stimulation for 10 s at 20 Hz. Fluorescence decay constant after stimulation was not different between the two probes (VAMP2-SEP 17.8 ± 1.5 s, VAMP2-SNAP-CFl 15.6 ± 1.0 s, $p = 0.27$, $n = 25$, Student's $t$-test). **c** Fluorescence images of hippocampal axons expressing VAMP2-SEP (left) and VAMP2-SNAP-VO (right) with the insets indicated with arrows (top). Time-lapse of the exocytosis events (bottom). Scale bar: 5 μm. **d** Average VAMP2-SEP (green) and VAMP2-SNAP-VO (magenta) fluorescence signals in response to field stimulation for 10 s at 20 Hz. Fluorescence decay after stimulation was not different between the two probes (VAMP2-SEP 13.2 ± 1.7 s; VAMP2-SNAP-VO 13.0 ± 1.5 s, $p = 0.51$, $n = 13$, Student's $t$-test). **e** Fluorescence images of hippocampal axons expressing VAMP2-SEP (left) and VAMP2-pHuji (right) with the insets indicated with arrows (top). Time-lapse of the exocytosis events (bottom). Scale bar: 5 μm. **f** Average VAMP2-SEP (green) and VAMP2-pHuji (magenta) fluorescence signals in response to field stimulation for 10 s at 20 Hz. Fluorescence decay constant after stimulation was 16.2 ± 0.8 s for SEP and 19.6 ± 0.9 s for pHuji ($n = 60$). Data are represented as mean ± s.e.m.

archetypical small-molecule pH sensor is fluorescein (Fl, **1**, Fig. 1b), which transitions between a highly fluorescent dianion (**$1^{2-}$**) and a less fluorescent monoanion (**$1^-$**) with a relatively low $pK_a$ value of 6.3 (Fig. 1c)[8]. Other unsuitable synthetic pH probes include the ratiometric seminapthorhodofluor (SNARF) dyes[9] that exhibit high background, as well as cyanine and rhodamine-based pH sensors that show the opposite pH sensitivity profile[10],[11].

We recently synthesized new derivatives of fluorescein (**1**) where the xanthene oxygen was replaced with a *gem*-dimethyl-carbon moiety. This work resulted in "carbofluorescein" (CFl, **2**, Fig. 1b)[12], and the difluorinated derivative "Virginia Orange" (VO, **3**, Fig. 1b)[13]. We discovered that this oxygen→carbon substitution elicited significant changes in photophysical and chemical properties of the fluorescein scaffold. Fl (**1**) exhibits $\lambda_{ex}/\lambda_{em} = 491$ nm/510 nm at high pH, whereas CFl (**2**) and VO (**3**) are red-shifted with $\lambda_{ex}/\lambda_{em} = 544$ nm/567 nm and 555 nm/581 nm, respectively. In addition to this bathochromic shift, the pH sensitivity of the dyes was markedly different. Fluorescein exhibits strong visible absorption at both pH 5.6 (vesicle pH) where the

monoanion **$1^-$** dominates, and pH 7.4 (extracellular pH) where the dianion form **$1^{2-}$** is prevalent—this can be observed by eye (Fig. 1c, d). In contrast, CFl (**2**) undergoes a cooperative transition between a highly colored dianion species (**$2^{2-}$**) and a colorless lactone form (**$2_{lactone}$**; Fig. 1c). This is also evident visually as a solution of CFl (**2**) is colorless at pH 5.6, but shows robust visible absorption at pH 7.4 (Fig. 1d). Fluorescence-based titrations (Fig. 1e) gave $pK_a$ values of 6.3 and Hill coefficient ($\eta_H$) value of 0.97 for fluorescein (**1**), consistent with previous reports[8]. CFl (**2**) and VO (**3**) displayed $pK_a$ values of 7.5 and 6.7, and $\eta_H$ values of 1.6 and 1.5, respectively. This cooperative transition likely stems from the altered lactone–quinoid equilibrium observed in the carbon-containing analogs of fluorescein and rhodamine dyes[12]. These pH sensitive red fluorophores have extinction coefficients and quantum yields similar to the pH sensitive red fluorescent proteins pHoran4 and pHuji[5] (Supplementary Table 1). Moreover, we tested the resistance of these fluorophores to photobleaching (Supplementary Fig. 1a). While CFl showed similar photobleaching rate as Fl ($\tau = 5.33$ s and $\tau = 5.35$ s), VO was two to three times more photostable ($\tau = 14.54$ s).

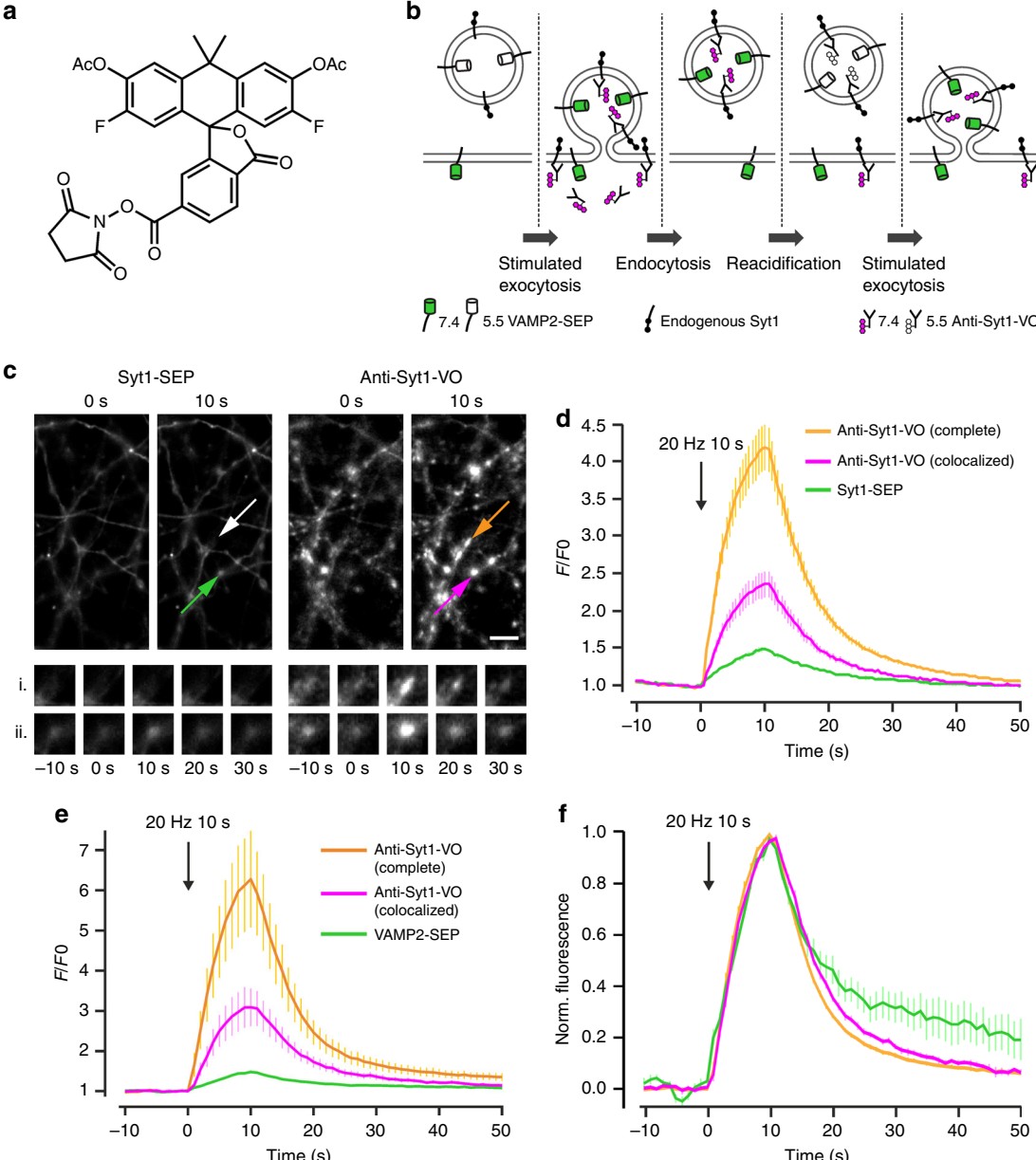

**Fig. 4** Detection of synaptic vesicle exocytosis with labeling of endogenous proteins. **a** Chemical structure of VO-NHS ester (6). **b** Protocol for staining with anti-Syt1-VO. **c** Fluorescence images of hippocampal axons expressing Syt1-SEP and stained with anti-Syt1-VO with the locations of vesicle fusion indicated with arrows (top); scale bar: 5 μm. Bottom, time-lapse images of the exocytosis events in boutons containing only anti-Syt1-VO (i) or containing both Syt1-SEP and anti-Syt1-VO (ii). **d** Average anti-Syt1-VO (magenta) and Syt1-SEP (green) fluorescence signals colocalized in the same boutons, and average anti-Syt1-VO fluorescence signals present in the complete neuronal population (orange) in response to field stimulation for 10 s at 20 Hz; $n = 41$. **e** Same as **d** for neurons transfected with VAMP2-SEP. Average anti-Syt1-VO (magenta) and VAMP2-SEP (green) fluorescence signals colocalized in the same boutons, and average anti-Syt1-VO fluorescence signals present in the complete neuronal population (orange); $n = 10$. **f** Average normalized traces corresponding to the recordings in **e**. Fluorescence decay after stimulation was $10.6 \pm 1.9$ s for VAMP2-SEP, $7.9 \pm 0.4$ s for anti-Syt1-VO co-localized with VAMP2-SEP, and $6.5 \pm 0.2$ for all boutons ($n = 10$ fields). Data are represented as mean $\pm$ s.e.m.

Therefore, the longer absorption and emission wavelengths, higher $pK_a$, resistance to photobleaching, and the cooperative colorless→colored transition upon increasing pH make both CFl and VO attractive scaffolds for building indicators to monitor synaptic vesicle fusion events.

To allow for specific labeling of expressed proteins, we prepared the SNAP-tag ligands attached to CFl (**4**) or VO (**5**) (Fig. 1f, Supplementary Fig. 1b). We tested the effects of protein conjugation on the properties of the dye by labeling SNAP-tag protein in vitro with CFl–SNAP-tag ligand **4** (Fig. 1g). We observed a shift in $pK_a$ to 7.3, and a decreased Hill coefficient ($\eta_H$ = 1.2; Fig. 1h). The active site of the SNAP-tag enzyme is flanked with two Lys, one Arg, and one His (PDB structure 3KZZ, DOI: 10.2210/pdb3kzz/pdb). The resulting Coulombic interaction between these positively charged amino acid residues and the CFl label most likely explains the decrease in $pK_a$ upon conjugation[8]. This polar surface might also stabilize the open form of the dye, resulting in the decreased cooperativity of the colored–colorless transition. Despite this lower $pK_a$ value and Hill coefficient, the fluorescence of the SNAP-tag-CFl conjugate is still completely suppressed at pH 5.6 (Fig. 1i).

**Detection of single exocytosis events in PC12 cells**. Next, we tested these SNAP-tag-based probes in living cells. Building on existing SEP-based constructs[1, 2], we designed several SNAP-tag fusion proteins: (i) SNAP-tag inserted within an intra-luminal loop of the vesicular acetylcholine transporter VAChT (VAChT-SNAP), and glutamate transporter VGluT1 (VGluT1-SNAP), and (ii) SNAP-tag protein attached to the luminal C-terminal side of the vesicle protein VAMP2 (VAMP2-SNAP). We first expressed VAChT-SNAP and VAMP2-SNAP in neuroendocrine PC12 cells. VAChT is targeted to small synaptic-like vesicles (SSLV) while VAMP2 is found in both SSLV and large dense core vesicles[14, 15]. We found that the propensity of the CFl and VO fluorophores to adopt the neutral lactone form (Fig. 1c, d) allows for efficient intracellular labeling (Fig. 2a) with SNAP-tag ligands **4** or **5** without the use of other masking groups (e.g., acetate esters), which are typically required for fluorescein-based compounds. To monitor exocytosis, we depolarized cells with stimulation buffer containing high $[K^+]$ and imaged single small vesicles as they fused with the plasma membrane using total internal reflection fluorescence (TIRF) microscopy. Cells expressing VAChT constructs displayed events at high frequency. Events detected in cells co-expressing VAChT-SEP and VAChT-SNAP (labeled with CFl ligand **4**) showed comparable fold increases in fluorescence at exocytosis ($2.19 \pm 0.07$ vs. $2.40 \pm 0.12$, mean $\pm$ s.e.m.) with similar decay kinetics in both the green and red channels (Fig. 2b, c). We also compared VAChT-SEP to VAChT-SNAP-VO (Fig. 2d, e) and VAChT-pHuji (Fig. 2f, g). Like the semisynthetic indicator from CFl ligand **4**, the VAChT-SNAP-VO derived from compound **5** also showed comparable performance to the SEP sensor ($2.65 \pm 0.10$ vs. $2.60 \pm 0.16$-fold increase; Fig. 2d, e). However, in PC12 cells the RFP-based VAChT-pHuji sensor showed lower relative performance when compared with VAChT-SEP under the same conditions ($2.01 \pm 0.05$ vs. $1.32 \pm 0.02$-fold increase; Fig. 2f, g) making events harder to detect with pHuji than with the other pH-sensitive proteins. We also observed individual fusion events using VAMP2-SEP or VAMP2-SNAP-CFl (Supplementary Fig. 2), albeit at low frequency, perhaps due to poor incorporation of this construct in PC12 cells.

**Monitoring exocytosis and recycling of synaptic vesicles**. We then tested these sensors in living neurons, focusing first on VAMP2-based constructs, which have been used extensively to follow SV exocytosis in neurons[1]. We co-transfected hippocampal neurons with VAMP2-SEP and either VAMP2-pHuji or VAMP2-SNAP incubated with CFl ligand **4** or Virginia Orange ligand **5**. For all the sensors, we observed a robust increase in fluorescence following electrical stimulation in fields covered with transfected axons, signaling SV exocytosis. The relative increase in fluorescence upon SV exocytosis was slightly higher for the SEP channel relative to the red-shifted fluorescent indicators, VAMP2-SNAP-CFl (Fig. 3a, b), VAMP2-SNAP-VO (Fig. 3c, d), and VAMP2-pHuji (Fig. 3e, f), which behaved similarly. The kinetics of decay, which tracks endocytosis and re-acidification of the vesicle, were similar for all four labels (Fig. 3b, d, f). We then tested whether the added CFl or VO could label the whole SV population efficiently. To do so, we normalized the fluorescence increase induced by stimulation with an application of a buffer containing 50 mM $NH_4Cl$, which quickly increases the intravesicular pH to extracellular pH[2]. We found that the proportion of SVs undergoing exocytosis estimated with this method was similar for all three red fluorescent probes (SNAP-CFl, SNAP-VO, and pHuji) and similar to SEP (Supplementary Fig. 3a–f). We also tested whether the semisynthetic pH sensor system could be used in multicolor imaging experiments with GFP-based indicators. We

co-transfected neurons with GCaMP6f[16] and VAMP2-SNAP, which we labeled with CFl ligand **4**. This allowed simultaneous imaging of both calcium ion transients and vesicle fusion in the same cell (Supplementary Fig. 3g–j). Finally, we tested the ability to monitor exocytosis by labeling another SV protein, VGluT1, which has been used previously with SEP to monitor SV exocytosis[17]. The three probes (VGluT1-SNAP-CFl, VGluT1-SNAP-VO, and VGluT1-pHuji) were able to report SV exocytosis (Supplementary Fig. 3k–m).

**Imaging of endogenous protein exo-/endocytosis cycling**. One interesting feature of small organic fluorophores is their ability to label not only genetically encoded domains, such as SNAP-tag, but also ligands and antibodies and hence endogenous proteins. To enable imaging of endogenous vesicular proteins, we labeled a monoclonal antibody that recognizes a luminal epitope of synaptotagmin 1 (Syt1), a SV protein, with $VOAc_2$-NHS ester (**6**, Supplementary Fig. 1b) followed by mild deprotection of the acetate esters using hydroxylamine (Fig. 4a). This antibody has previously been used to detect endogenous Syt1 present on the plasma membrane after exocytosis in active synapses[18]. To mark vesicular Syt1, we incubated neurons with this antibody-VO conjugate (10 nM) for 3 h in stimulation buffer, followed by extensive washing to remove the unbound antibodies (Fig. 4b). The antibody labeling was done in neurons transfected with Syt1-SEP or VAMP2-SEP to compare the performance of this labeling technique with the overexpressed, genetically encoded GFP-based pH sensors. Electrical stimulation evoked a robust increase in fluorescence in axons transfected with Syt1-SEP (Fig. 4c, d) or VAMP2-SEP (Fig. 4e), and in axons of untransfected neurons without detectable SEP. We found that the decay kinetics after stimulation was faster for the VO-antibody conjugate ($8.8 \pm 0.5$ s) than the Syt1-SEP ($12.2 \pm 0.9$ s, $n = 41$; paired $t$-test $p < 0.0001$, Fig. 4d) and VAMP2-SEP (Fig. 4f), suggesting a difference in overexpressed vs. endogenous protein behavior after SV exocytosis[19]. Remarkably, the fluorescence transients were substantially higher in untransfected than in transfected neurons (Fig. 4d, e), perhaps stemming from steric hindrance of the overexpressed SEP proteins or through quenching of the two fluorophores.

## Discussion

We have developed new "semisynthetic" pH-sensitive proteins that allow for the imaging of synaptic vesicle fusion events in living cells. This sensor system combines the highly tunable properties of small-molecule fluorophores with the specificity of self-labeling tags or antibodies. The SNAP-tag-based system constitutes the first genetically encoded long-wavelength pH sensor with similar or better performance than SEP in different cell types. The antibody-based pH sensor allows for imaging of vesicle fusion events without the need for overexpression of sensor proteins. Addition of other self-labeling or epitope tags by genome editing methods could allow cell- and protein-specific labeling without the need for overexpression[20]. Future improvements of both the protein and the dye within this semi-synthetic scaffold should further enable imaging of this key biological process in increasingly complex systems.

## Methods

**General organic synthesis methods**. Commercial reagents were obtained from reputable suppliers and used as received. All solvents were purchased in septum-sealed bottles stored under an inert atmosphere. All reactions were sealed with septa through which a nitrogen atmosphere was introduced unless otherwise noted. Reactions were conducted in round-bottomed flasks or septum-capped crimp-top vials containing Teflon-coated magnetic stir bars. Heating of reactions was accomplished with a silicon oil bath or an aluminum reaction block on top of a stirring hotplate equipped with an electronic contact thermometer to maintain the indicated temperatures.

Reactions were monitored by thin layer chromatography (TLC) on precoated TLC glass plates (silica gel 60 F$_{254}$, 250 μm thickness) or by LC/MS (Phenomenex Kinetex 2.1 mm × 30 mm 2.6 μm C18 column; 5 μL injection; 5–98% MeCN/H$_2$O, linear gradient, with constant 0.1% v/v HCO$_2$H additive; 6 min run; 0.5 mL/min flow; ESI; positive ion mode). TLC chromatograms were visualized by UV illumination or developed with anisaldehyde, ceric ammonium molybdate, or KMnO$_4$ stain. Reaction products were purified by flash chromatography on an automated purification system using pre-packed silica gel columns or by preparative HPLC (Phenomenex Gemini–NX 30 × 150 mm 5 μm C18 column). Analytical HPLC analysis was performed with an Agilent Eclipse XDB 4.6 × 150 mm 5 μm C18 column under the indicated conditions. High-resolution mass spectrometry was performed by the Mass Spectrometry Center in the Department of Medicinal Chemistry at the University of Washington and the High Resolution Mass Spectrometry Facility at the University of Iowa.

NMR spectra were recorded on a 400 MHz spectrometer. $^1$H and $^{13}$C chemical shifts (δ) were referenced to TMS or residual solvent peaks, and $^{19}$F chemical shifts (δ) were referenced to CFCl$_3$. Data for $^1$H NMR spectra are reported as follows: chemical shift (δ ppm), multiplicity (s = singlet, d = doublet, t = triplet, q = quartet, dd = doublet of doublets, m = multiplet), coupling constant (Hz), integration. Data for $^{13}$C NMR spectra are reported by chemical shift (δ ppm) with hydrogen multiplicity (C, CH, CH$_2$, CH$_3$) information obtained from DEPT spectra. Original spectra for all reported $^1$H and $^{13}$C NMR data are given in Supplementary Figs. 4–26. Since the amount of compound **5** was not enough for $^{13}$C NMR, only LC/MS analysis was performed (Supplementary Fig. 27).

**Synthesis of TBS$_2$-CFl-6-CO$_2$t-Bu (10).** A vial was charged with di-tert-butyl 2-bromoterephthalate (**9**; 1.48 g, 4.14 mmol, 2 eq), sealed, and flushed with nitrogen. After dissolving the bromide in THF (7 mL) and cooling the reaction to −15 °C, iPrMgCl•LiCl (1.3 M in THF, 3.19 mL, 4.14 mmol, 2 eq) was added. The reaction was warmed to −10 °C and stirred for 4 h. A solution of 3,6-bis((tert-butyldimethylsilyl)oxy)-10,10-dimethylanthracen-9(10H)-one$^{12}$ (**7**; 1.00 g, 2.07 mmol) in THF (4 mL) was then added dropwise. The reaction mixture was warmed to room temperature and stirred for 2 h. It was subsequently quenched with saturated NH$_4$Cl, diluted with water, and extracted with EtOAc (2×). The combined organics were washed with brine, dried over anhydrous MgSO$_4$, filtered, and evaporated. Silica gel chromatography (0–10% Et$_2$O/hexanes, linear gradient) provided 245 mg (17%) of **10** as a colorless solid. $^1$H NMR (CDCl$_3$, 400 MHz) δ 8.16 (dd, J = 8.0, 1.3 Hz, 1H), 8.02 (dd, J = 8.0, 0.6 Hz, 1H), 7.63–7.59 (m, 1H), 7.09–7.05 (m, 2H), 6.64–6.57 (m, 4H), 1.81 (s, 3H), 1.72 (s, 3H), 1.54 (s, 9H), 0.99 (s, 18H), 0.22 (s, 12H); $^{13}$C NMR (CDCl$_3$, 101 MHz) δ 169.9 (C), 164.4 (C), 156.5 (C), 155.5 (C), 147.0 (C), 138.1 (C), 130.3 (CH), 129.7 (C), 129.3 (CH), 125.1 (CH), 125.0 (CH), 124.0 (C), 119.2 (CH), 117.8 (CH), 87.0 (C), 82.5 (C), 38.2 (C), 35.0 (CH$_3$), 33.2 (CH$_3$), 28.2 (CH$_3$), 25.8 (CH$_3$), 18.4 (C), −4.17 (CH$_3$), −4.19 (CH$_3$); HRMS (ESI) calcd for C$_{40}$H$_{55}$O$_6$Si$_2$ [M + H]$^+$ 687.3537, found 687.3533.

**Synthesis of TBS$_2$-VO-6-CO$_2$t-Bu (11).** A vial was charged with di-tert-butyl 2-bromoterephthalate (**9**; 1.03 g, 2.89 mmol, 1.5 eq), sealed, and flushed with nitrogen. After dissolving the bromide in THF (5 mL) and cooling the reaction to −50 °C, i-PrMgCl•LiCl (1.3 M in THF, 2.22 mL, 2.89 mmol, 1.5 eq) was added. The reaction was stirred at −40 °C for 4 h. A solution of 3,6-bis((tert-butyldimethylsilyl)oxy)−2,7-difluoro-10,10-dimethylanthracen-9(10H)-one$^{13}$ (**8**; 1.00 g, 1.93 mmol) in THF (5 mL) was then added dropwise. The reaction mixture was warmed to room temperature and stirred for 2 h. It was subsequently quenched with saturated NH$_4$Cl, diluted with water, and extracted with EtOAc (2×). The combined organics were washed with brine, dried over anhydrous MgSO$_4$, filtered, and evaporated. Silica gel chromatography (0–10% Et$_2$O/hexanes, linear gradient) provided 439 mg (31%) of **11** as a white solid. $^1$H NMR (CDCl$_3$, 400 MHz) δ 8.20 (dd, J = 8.0, 1.3 Hz, 1H), 8.05 (dd, J = 8.0, 0.6 Hz, 1H), 7.65–7.60 (m, 1 H), 7.12 (d, J = 8.3 Hz, 2H), 6.36 (d, J = 11.3 Hz, 2H), 1.77 (s, 3H), 1.68 (s, 3H), 1.56 (s, 9H), 1.01 (s, 18H), 0.21 (s, 12H); $^{19}$F NMR (CDCl$_3$, 376 MHz) δ −133.17−−133.27 (m); $^{13}$C NMR (CDCl$_3$, 101 MHz) δ 169.3 (C), 164.2 (C), 154.6 (C), 152.8 (d, $^1J_{CF}$ = 245.8 Hz, C), 144.6 (d, $^2J_{CF}$ = 12.7 Hz, C), 141.6 (d, $^4J_{CF}$ = 3.4 Hz, C), 138.5 (C), 130.9 (CH), 129.2 (C), 125.5 (CH), 124.8 (CH), 124.4 (d, $^3J_{CF}$ = 5.5 Hz, CH), 120.3 (d, $^3J_{CF}$ = 1.9 Hz, CH), 115.0 (d, $^2J_{CF}$ = 20.1 Hz, CH), 85.7 (C), 82.7 (C), 37.6 (C), 35.1 (CH$_3$), 33.6 (CH$_3$), 28.2 (CH$_3$), 25.7 (CH$_3$), 18.5 (C), −4.5 (CH$_3$); HRMS (ESI) calcd for C$_{40}$H$_{53}$F$_2$O$_6$Si$_2$ [M + H]$^+$ 723.3343, found 723.3352.

**CFl-6-CO$_2$t-Bu (12).** A solution of **10** (215 mg, 0.313 mmol) in THF (5 mL) was cooled to 0 °C, and TBAF (1.0 M in THF, 1.25 mL, 1.25 mmol, 4 eq) was added. The reaction was warmed to room temperature and stirred for 1 h. It was subsequently diluted with saturated NH$_4$Cl and extracted with EtOAc (2×). The organic extracts were washed with brine, dried over anhydrous MgSO$_4$, filtered, and evaporated. Flash chromatography (10–100% EtOAc/hexanes, linear gradient) yielded **12** (137 mg, 95%) as a pale orange solid. $^1$H NMR (DMSO-$d_6$, 400 MHz) δ 9.74 (s, 2H), 8.14 (dd, J = 8.0, 1.2 Hz, 1H), 8.10 (dd, J = 8.0, 0.6 Hz, 1H), 7.40–7.36 (m, 1H), 7.10 (d, J = 2.4 Hz, 2H), 6.63 (dd, J = 8.6, 2.4 Hz, 2H), 6.51 (d, J = 8.6 Hz, 2H), 1.74 (s, 3H), 1.65 (s, 3H), 1.48 (s, 9H); $^{13}$C NMR (DMSO-$d_6$, 101 MHz) δ 168.8 (C), 163.5 (C), 158.1 (C), 155.2 (C), 146.5 (C), 137.3 (C), 130.1 (CH), 128.9 (C), 128.8 (CH), 125.4 (CH), 123.4 (CH), 121.0 (C), 115.1 (CH),

112.7 (CH), 86.5 (C), 82.2 (C), 37.6 (C), 34.3 (CH$_3$), 33.3 (CH$_3$), 27.5 (CH$_3$); HRMS (ESI) calcd for C$_{28}$H$_{27}$O$_6$ [M+H]$^+$ 459.1802, found 459.1818.

**VO-6-CO$_2$t-Bu (13).** To a solution of **11** (300 mg, 0.415 mmol) in THF (5 mL) was added TBAF (1.0 M in THF, 1.66 mL, 1.66 mmol, 4 eq). The reaction was stirred at room temperature for 30 min. It was subsequently acidified with 1 N HCl, diluted with water, and extracted with EtOAc (2×). The organic extracts were dried over anhydrous MgSO$_4$, filtered, and evaporated. Flash chromatography (0–30% EtOAc/toluene, linear gradient) afforded **13** (201 mg, 98%) as a yellow-orange solid. $^1$H NMR (DMSO-$d_6$, 400 MHz) δ 10.28 (s, 2H), 8.16 (dd, J = 8.0, 1.3 Hz, 1H), 8.11 (dd, J = 8.0, 0.7 Hz, 1H), 7.44–7.40 (m, 1H), 7.29 (d, $^4J_{HF}$ = 8.8 Hz, 2H), 6.41 (d, $^3J_{HF}$ = 11.9 Hz, 2H), 1.73 (s, 3H), 1.63 (s, 3H), 1.48 (s, 9H); $^{19}$F NMR (DMSO-$d_6$, 376 MHz) δ −136.55 (dd, $J_{FH}$ = 11.8, 8.9 Hz); $^{13}$C NMR (DMSO-$d_6$, 101 MHz) δ 168.4 (C), 163.5 (C), 154.2 (C), 149.8 (d, $^1J_{CF}$ = 242.7 Hz, C), 146.2 (d, $^2J_{CF}$ = 12.5 Hz, C), 141.5 (d, $^4J_{CF}$ = 3.0 Hz, C), 137.5 (C), 130.5 (CH), 128.5 (C), 125.9 (CH), 123.2 (CH), 121.3 (d, $^3J_{CF}$ = 5.3 Hz, C), 115.7 (d, $^3J_{CF}$ = 2.7 Hz, CH), 113.9 (d, $^2J_{CF}$ = 18.8 Hz, CH), 85.0 (C), 82.2 (C), 37.1 (C), 34.2 (CH$_3$), 33.7 (CH$_3$), 27.5 (CH$_3$); HRMS (ESI) calcd for C$_{28}$H$_{25}$F$_2$O$_6$ [M + H]$^+$ 495.1614, found 495.1625.

**CFl-6-CO$_2$H (14).** Ester **12** (75 mg, 0.164 mmol) was taken up in CH$_2$Cl$_2$ (3 mL), and trifluoroacetic acid (0.6 mL) was added. The reaction was stirred at room temperature for 6 h. Toluene (3 mL) was added; the reaction mixture was concentrated to dryness and then azeotroped with MeOH three times to provide **14** as a red solid (65 mg, 99%). Analytical HPLC and NMR indicated that the material was >95% pure and did not require further purification. $^1$H NMR (DMSO-$d_6$, 400 MHz) δ 13.57 (s, 1H), 9.74 (s, 2H), 8.17 (dd, J = 8.0, 1.3 Hz, 1H), 8.09 (dd, J = 8.0, 0.7 Hz, 1H), 7.44 (dd, J = 1.2, 0.8 Hz, 1H), 7.09 (d, J = 2.5 Hz, 2H), 6.62 (dd, J = 8.6, 2.5 Hz, 2H), 6.50 (d, J = 8.6 Hz, 2H), 1.74 (s, 3H), 1.64 (s, 3H); $^{13}$C NMR (DMSO-$d_6$, 101 MHz) δ 168.9 (C), 166.0 (C), 158.1 (C), 155.2 (C), 146.6 (C), 137.1 (C), 130.3 (CH), 128.92 (C), 128.90 (CH), 125.3 (CH), 123.9 (CH), 121.0 (C), 115.0 (CH), 112.7 (CH), 86.5 (C), 37.6 (C), 34.4 (CH$_3$), 33.2 (CH$_3$); HRMS (ESI) calcd for C$_{24}$H$_{17}$O$_6$ [M−H]$^-$ 401.1031, found 401.1037.

**VO-6-CO$_2$H (15).** Ester **13** (185 mg, 0.374 mmol) was taken up in CH$_2$Cl$_2$ (5 mL), and trifluoroacetic acid (1 mL) was added. The reaction was stirred at room temperature for 8 h. Toluene (3 mL) was added; the reaction mixture was concentrated to dryness and then azeotroped with MeOH three times to provide acid **15** as an orange solid (158 mg, 96%). Analytical HPLC and NMR indicated that the material was >95% pure and did not require further purification. $^1$H NMR (DMSO-$d_6$, 400 MHz) δ 13.58 (s, 1H), 10.27 (s, 2H), 8.20 (dd, J = 8.0, 1.3 Hz, 1H), 8.11 (dd, J = 8.0, 0.5 Hz, 1H), 7.50–7.46 (m, 1H), 7.29 (d, $^4J_{HF}$ = 8.8 Hz, 2H), 6.40 (d, $^3J_{HF}$ = 11.9 Hz, 2H), 1.73 (s, 3H), 1.63 (s, 3H); $^{19}$F NMR (DMSO-$d_6$, 376 MHz) δ −136.63 (dd, $J_{FH}$ = 11.8, 8.9 Hz); $^{13}$C NMR (DMSO-$d_6$, 101 MHz) δ 168.5 (C), 165.9 (C), 154.2 (C), 149.8 (d, $^2J_{CF}$ = 242.6 Hz, C), 146.2 (d, $^2J_{CF}$ = 12.5 Hz, C), 141.6 (d, $^4J_{CF}$ = 3.0 Hz, C), 137.3 (C), 130.7 (CH), 128.6 (C), 125.9 (CH), 123.6 (CH), 121.4 (d, $^3J_{CF}$ = 5.2 Hz, C), 115.6 (d, $^3J_{CF}$ = 2.8 Hz, CH), 114.0 (d, $^2J_{CF}$ = 18.9 Hz, CH), 85.1 (C), 37.2 (C), 34.4 (CH$_3$), 33.5 (CH$_3$); HRMS (ESI) calcd for C$_{24}$H$_{15}$F$_2$O$_6$ [M−H]$^-$ 437.0842, found 437.0853.

**Ac$_2$-CFl-6-CO$_2$H (16).** Dye **14** (90 mg, 0.224 mmol), pyridine (90 μL, 1.12 mmol, 5 eq), and acetic anhydride (2 mL) were combined in a vial and stirred at 80 °C for 1 h. The reaction was cooled to room temperature, concentrated in vacuo, diluted with 1 N HCl, and extracted with EtOAc (2×). The organic layers were washed with brine, dried over anhydrous MgSO$_4$, filtered, deposited onto silica gel, and evaporated. The residue was purified by flash chromatography (20–100% EtOAc/hexanes, linear gradient, with constant 1% v/v AcOH; dry load with silica gel) to afford 102 mg (94%) of **16** as a white foam. $^1$H NMR (CDCl$_3$, 400 MHz) δ 8.29 (dd, J = 8.0, 1.2 Hz, 1H), 8.11 (dd, J = 8.0, 0.5 Hz, 1H), 7.75–7.71 (m, 1H), 7.38 (d, J = 2.3 Hz, 2H), 6.91 (dd, J = 8.7, 2.4 Hz, 2H), 6.75 (d, J = 8.7 Hz, 2H), 2.31 (s, 6H), 1.86 (s, 3H), 1.75 (s, 3H); $^{13}$C NMR (CDCl$_3$, 101 MHz) δ 169.6 (C), 169.23 (C), 169.15 (C), 154.9 (C), 151.6 (C), 146.8 (C), 135.6 (C), 131.5 (CH), 130.4 (C), 129.2 (CH), 128.2 (C), 125.8 (C), 125.7 (CH), 120.8 (CH), 119.7 (CH), 85.8 (C), 38.7 (C), 35.0 (CH$_3$), 33.0 (CH$_3$), 21.3 (CH$_3$); HRMS (ESI) calcd for C$_{28}$H$_{21}$O$_8$ [M−H]$^-$ 485.1242, found 485.1235.

**Ac$_2$-VO-6-CO$_2$H (17).** Dye **15** (125 mg, 0.285 mmol), pyridine (115 μL, 1.43 mmol, 5 eq), and acetic anhydride (2.5 mL) were combined in a vial and stirred at 80 °C for 1 h. The reaction was cooled to room temperature, diluted with 1 N HCl, and extracted with EtOAc (2×). The organic layers were washed with brine, dried over anhydrous MgSO$_4$, filtered, and evaporated. The residue was purified by flash chromatography (10–100% EtOAc/hexanes, linear gradient, with constant 1% v/v AcOH) to afford 128 mg (86%) of **17** as an off-white solid. $^1$H NMR (DMSO-$d_6$, 400 MHz) δ 13.62 (s, 1H), 8.23 (dd, J = 8.0, 1.3 Hz, 1H), 8.16 (dd, J = 8.0, 0.7 Hz, 1H), 7.84 (d, $^4J_{HF}$ = 7.7 Hz, 2H), 7.60–7.55 (m, 1H), 6.76 (d, $^3J_{HF}$ = 11.0 Hz, 2H), 2.34 (s, 6H), 1.81 (s, 3H), 1.70 (s, 3H); $^{19}$F NMR (DMSO-$d_6$, 376 MHz) δ −129.33 (dd, $J_{FH}$ = 11.0, 7.7 Hz); $^{13}$C NMR (DMSO-$d_6$, 101 MHz) δ 168.1 (C), 168.0 (C), 165.9 (C), 152.2 (d, $^1J_{CF}$ = 248.1 Hz, C), 141.6 (d, $^4J_{CF}$ = 3.3 Hz, C), 138.9 (d, $^2J_{CF}$ = 13.2 Hz, C), 137.7 (C), 131.1 (CH), 129.2 (d, $^3J_{CF}$ = 5.9 Hz, C), 128.3 (C), 126.4 (CH), 123.6 (CH), 123.2 (CH), 114.6 (d,

$^2J_{CF}$ = 19.7 Hz, CH), 83.6 (C), 37.8 (C), 33.9 (CH$_3$), 33.4 (CH$_3$), 20.2 (CH$_3$); HRMS (ESI) calcd for C$_{28}$H$_{19}$F$_2$O$_8$ [M−H]$^-$ 521.1053, found 521.1050.

**Ac$_2$-CFl-6-NHS (18)**. To a solution of acid **16** (30 mg, 0.0617 mmol) and TSTU (37 mg, 0.123 mmol, 2 eq) in DMF (1.5 mL) was added DIEA (54 µL, 0.309 mmol, 5 eq). The reaction was stirred at room temperature for 2. It was subsequently diluted with 10% w/v citric acid and extracted with EtOAc (2×). The combined organic extracts were washed with brine, dried over anhydrous MgSO$_4$, filtered, deposited onto silica gel, and concentrated in vacuo. Silica gel chromatography (25–100% EtOAc/hexanes, linear gradient; dry load with silica gel) yielded 30 mg (83%) of **18** as an off-white solid. $^1$H NMR (CDCl$_3$, 400 MHz) δ 8.37 (dd, J = 8.0, 1.4 Hz, 1H), 8.17 (dd, J = 8.0, 0.7 Hz, 1H), 7.80 (dd, J = 1.3, 0.8 Hz, 1H), 7.41 (d, J = 2.3 Hz, 2H), 6.94 (dd, J = 8.7, 2.4 Hz, 2H), 6.73 (d, J = 8.7 Hz, 2H), 2.87 (s, 4H), 2.31 (s, 6H), 1.86 (s, 3H), 1.75 (s, 3H); $^{13}$C NMR (CDCl$_3$, 101 MHz) δ 169.1 (C), 168.8 (C), 168.5 (C), 160.7 (C), 154.8 (C), 151.8 (C), 147.0 (C), 131.8 (CH), 131.7 (C), 131.4 (C), 129.3 (CH), 127.9 (C), 126.3 (CH), 126.1 (CH), 120.9 (CH), 119.7 (CH), 86.0 (C), 38.8 (C), 35.3 (CH$_3$), 32.6 (CH$_3$), 25.8 (CH$_2$), 21.3 (CH$_3$); HRMS (ESI) calcd for C$_{32}$H$_{26}$NO$_{10}$ [M + H]$^+$ 584.1551, found 584.1566.

**Ac$_2$-VO-6-NHS (6)**. To a solution of acid **17** (53 mg, 0.101 mmol) and TSTU (61 mg, 0.203 mmol, 2 eq) in DMF (2 mL) was added DIEA (44 µL, 0.254 mmol, 2.5 eq). The reaction was stirred at room temperature for 2 h. It was subsequently diluted with 10% w/v citric acid and extracted with EtOAc (2×). The combined organic extracts were washed with brine, dried over anhydrous MgSO$_4$, filtered, and concentrated in vacuo. Silica gel chromatography (25–100% EtOAc/hexanes, linear gradient) yielded 52 mg (83%) of **6** as a white solid. $^1$H NMR (CDCl$_3$, 400 MHz) δ 8.41 (dd, J = 8.0, 1.3 Hz, 1H), 8.20 (dd, J = 8.0, 0.7 Hz, 1H), 7.83–7.78 (m, 1H), 7.44 (d, $^4J_{HF}$ = 7.3 Hz, 2H), 6.50 (d, $^3J_{HF}$ = 10.5 Hz, 2H), 2.88 (s, 4H), 2.34 (s, 6H), 1.82 (s, 3H), 1.73 (s, 3H); $^{19}$F NMR (CDCl$_3$, 376 MHz) δ −128.60 (dd, $J_{FH}$ = 10.4, 7.4 Hz); $^{13}$C NMR (CDCl$_3$, 101 MHz) δ 168.7 (C), 167.9 (C), 160.6 (C), 153.9 (C), 152.8 (d, $^1J_{CF}$ = 251.4 Hz, C), 141.6 (d, $^4J_{CF}$ = 3.6 Hz, C), 139.7 (d, $^2J_{CF}$ = 13.2 Hz, C), 132.3 (CH), 131.8 (C), 130.9 (C), 129.0 (d, $^3J_{CF}$ = 5.8 Hz, C), 126.5 (CH), 126.0 (CH), 122.4 (CH), 115.41 (d, $^2J_{CF}$ = 19.9 Hz, CH), 84.8 (C), 38.2 (C), 35.2 (CH$_3$), 33.4 (CH$_3$), 25.8 (CH$_2$), 20.6 (CH$_3$); HRMS (ESI) calcd for C$_{32}$H$_{24}$F$_2$NO$_{10}$ [M + H]$^+$ 620.1363, found 620.1370.

**CFl-SNAP-tag ligand (4)**. Ester **18** (30 mg, 51.4 µmol) and 6-((4-(aminomethyl)benzyl)oxy)-9H-purin-2-amine (BG-NH$_2$, **19**; 18.1 mg, 66.8 µmol, 1.3 eq) were combined in DMF (2 mL). DIEA (17.9 µL, 103 µmol, 2 eq) was added, and the reaction was stirred at room temperature for 2 h. MeOH (1 mL) and 1 N NaOH (200 µL) were then added, and the mixture was stirred at room temperature for an additional 2 h. The reaction was acidified with 1 N HCl (400 µL), diluted with water, and extracted with 15% i-PrOH/CHCl$_3$ (2×). The combined organic extracts were dried over anhydrous MgSO$_4$, filtered, deposited onto Celite, and concentrated in vacuo. The crude product was purified by flash chromatography on silica gel (0–10% MeOH/EtOAc, linear gradient; dry load with Celite) to provide ligand **4** (30 mg, 89%) as an orange solid. $^1$H NMR (MeOD, 400 MHz) δ 8.08 (dd, J = 8.0, 1.4 Hz, 1H), 8.05 (dd, J = 8.0, 0.8 Hz, 1H), 7.82 (s, 1H), 7.48 (dd, J = 1.3, 0.8 Hz, 1H), 7.43 (d, J = 8.2 Hz, 2H), 7.28 (d, J = 8.2 Hz, 2H), 7.10 (d, J = 2.4 Hz, 2H), 6.60 (dd, J = 8.7, 2.5 Hz, 2H), 6.53 (d, J = 8.6 Hz, 2H), 5.50 (s, 2H), 4.48 (s, 2H), 1.80 (s, 3H), 1.70 (s, 3H); $^{13}$C NMR (MeOD, 101 MHz) δ 162.3 (C), 158.8 (C), 152.1 (C), 151.8 (C), 150.0 (C), 147.8 (C), 139.1 (C), 132.7 (C), 130.5 (C), 130.1 (C), 127.4 (C), 120.8 (CH), 120.5 (C), 120.3 (CH), 120.1 (CH), 119.2 (CH), 116.6 (CH), 114.3 (CH), 113.6 (C), 106.4 (CH), 104.3 (CH), 79.7 (C), 59.1 (CH$_2$), 35.0 (CH$_2$), 29.7 (C), 25.9 (CH$_3$), 23.8 (CH$_3$); analytical HPLC: >99% purity (5 µL injection; 10–95% MeCN/H$_2$O, linear gradient, with constant 0.1% v/v TFA additive; 20 min run; 1 mL/min flow; ESI; positive ion mode; detection at 254 nm); HRMS (ESI) calcd for C$_{37}$H$_{31}$N$_6$O$_6$ [M + H]$^+$ 655.2300, found 655.2306.

**VO-SNAP-tag ligand (5)**. Ester **6** (30 mg, 48.4 µmol) and 6-((4-(aminomethyl)benzyl)oxy)-9H-purin-2-amine (BG-NH$_2$, **19**; 17.0 mg, 63.0 µmol, 1.3 eq) were combined in DMF (2 mL), and DIEA (16.9 µL, 96.8 µmol, 2 eq) was added. After stirring the reaction for 1 h at room temperature, MeOH (1 mL) and 1 M NaOH (150 µL) were added. The mixture was stirred at room temperature for an additional 1 h. It was then acidified with 1 M HCl (300 µL), diluted with water, and extracted with 15% i-PrOH/CHCl$_3$ (2×). The combined organic extracts were dried over anhydrous MgSO$_4$, filtered, and concentrated in vacuo. The crude product was purified by silica gel chromatography (0–10% MeOH/EtOAc, linear gradient) to provide 22.1 mg (66%) of ligand **5** as a pink solid. $^1$H NMR (DMSO-d$_6$, 400 MHz) δ 12.38 (s, 1H), 10.26 (s, 2H), 9.25 (t, J = 5.8 Hz, 1H), 8.18 (dd, J = 8.0, 1.3 Hz, 1H), 8.08 (d, J = 7.9 Hz, 1H), 7.78 (s, 1H), 7.54–7.48 (m, 1H), 7.42 (d, J = 8.1 Hz, 2H), 7.29 (d, J = 8.1 Hz, 2H), 7.28 (d, $^4J_{HF}$ = 8.9 Hz, 2H), 6.35 (d, $^3J_{HF}$ = 11.9 Hz, 2H), 6.24 (s, 2H), 5.43 (s, 2H), 4.42 (d, J = 5.7 Hz, 2H), 1.73 (s, 3H), 1.62 (s, 3H); $^{19}$F NMR (DMSO-d$_6$, 376 MHz) δ −136.71 (dd, $J_{FH}$ = 11.8, 8.9 Hz); Analytical HPLC: 98.6% purity (5 µL injection; 10–95% CH$_3$CN/H$_2$O, linear gradient, with constant 0.1% v/v TFA additive; 20 min run; 1 mL/min flow; ESI; positive ion mode; UV detection at 254 nm); HRMS (ESI) calcd for C$_{37}$H$_{29}$F$_2$N$_6$O$_6$ [M + H]$^+$ 691.2111, found 691.2127.

**UV-vis and fluorescence spectroscopy**. Spectroscopy was performed using 1-cm path length, 3.5-mL quartz cuvettes or 100-µL quartz microcuvettes Starna Cells. All measurements were taken at ambient temperature (~ 22 °C). Absorption spectra were recorded on a Cary Model 100 spectrometer (Varian), and fluorescence spectra were recorded on a Cary Eclipse fluorometer (Varian). The pK$_a$ values for compounds **1**, **2**, and **3** were determined in buffers containing 150 mM NaCl and 10 mM buffer. The following buffer systems were used: citrate (pH 4.0–6.2); phosphate (pH 5.8–8.0); tris (pH 7.8–9.0); carbonate (pH 9.2–10.0). Fluorescence values were read on 500 nM samples (n = 3) and fitted to a sigmoidal dose response curve (variable slope) using GraphPad Prism software. Samples for visual inspection containined 5 µM **1** or **2** in citrate (pH 5.6) or phosphate (pH 7.4) buffer.

**Measurement of fluorophore bleaching**. Solutions of 1 µM fluorescein (**1**), carbofluorescein (**2**), or Virginia Orange (**3**) were prepared in 50 mM sodium borate, pH 9.7. To create aqueous microdroplets, a sample of these solutions was added to 1-octanol in 1:9 ratio and this mixture was vortexed briefly. The resulting emulsion was placed on a glass slide and fitted with a coverslip. Fluorophore bleaching was accomplished by illuminating an entire isolated microdroplet of aqueous dye solution using an upright microscope (Zeiss Axio Observer Z2) and a 20×, NA 0.8 objective. Fluorescein (**1**) was bleached under 488 nm (Sapphire 488, Coherent) laser excitation at 4.3 mW power (Intensity = 3.25 W/cm$^2$) and the emission was collected using a 550BP88 filter. Carbofluorescein (**2**) and Virginia Orange (**3**) fluorescence were measured using a mercury lamp (X-Cite, Series 120-Q) with 550BP25 excitation at 6.3 mW power (intensity 4.76 W/cm$^2$); emission was collected using a 625BP90 filter. Fluorescence was detected by a fiber coupled Avalanche photodiode (SPQM-AQRH14, Pacer).

**Protein chemistry**. SNAP-tag protein (NEB) was labeled with excess CFl SNAP-tag ligand (**4**) in PBS containing 1 mM DTT. The protein concentrate was purified and concentrated using a Ziba spin desalting column (7K MWCO; ThermoFisher). The protein sample was analyzed by gel electrophoresis using a NuPAGE 4–12% Bis-Tris polyacrylamide gel (ThermoFisher) and imaged using a Typhoon Trio + scanner (GE Healthcare); the gel was also stained with Coomassie Plus (ThermoFisher) and compared to a Mark 12 protein standard ladder (ThermoFisher). Absorbance measurements were taken in citrate (pH 5.6) or phosphate (pH 7.4) buffers described above.

**Plasmid constructs**. VAMP2-SEP and synaptotagmin1-SEP were kind gifts from Jürgen Klingauf. GCaMP6f was obtained from Addgene (#40755). VaChT-SNAP and VaChT-pHuji were created by replacing the open reading frame of SEP for either SNAP-tag (New England Biolabs) or pHuji[5] in the VAChT-SEP construct[15]. VGluT1-SNAP and VGluT1-pHuji were created by replacing the open reading frame of SEP for either SNAP-tag (New England Biolabs) or pHuji[5] in the VGluT1-SEP construct[17]. VAMP2-pHuji was created by swapping the SEP gene for pHuji in VAMP2-SEP sensor construct[2].

**Cell culture and transfections**. PC12 cells (obtained from Dr. Rae Nishi, The Molecular Biology Labs) were grown in DMEM containing 4 mM L-glutamine, supplemented with 5% fetal bovine serum, 5% horse serum, and 1% penicillin/streptomycin, at 37 °C in 5% CO$_2$. Cells were plated onto 20-mm round poly-D-lysine-coated glass cover-slips and transfected ~24–48 h later with 1 µg VAChT-SEP, VAChT-SNAP, VAChT-pHuji, VAMP2-SEP or VAMP2-SNAP plasmids using Lipofectamine 2000 (Invitrogen) following the manufacturer's protocol. Cells were imaged 24 h after transfection.

Dissociated hippocampal neurons from E18 rat embryos of either sex were prepared at a density of 300,000 cells per dish on poly-D-lysine-coated coverslips, and maintained at 37 °C and 5% CO$_2$ in Neurobasal medium supplemented with 2 mM glutamine and 1× NeuroCult SM1 Neuronal supplement (STEMCELL Technologies)[21]. Transfection of VAMP2-SEP, VAMP2-pHuji, VAMP2-SNAP, VGluT1-SEP, VGluT1-pHuji, VGluT1-SNAP, GCaMP6f, and Syt1-SEP was performed at 6 days in vitro (DIV) by a modified calcium phosphate transfection procedure[22]. Experiments were carried out at 20–25 DIV.

**Antibody labeling**. Mouse monoclonal antibodies against the luminal part of Synaptotagmin 1 (100 µg, Synaptic Systems 105 311) were coupled in carbonate buffer (pH 9) to 3.4 eq VOAc$_2$-NHS ester (**6**; solubilized in DMSO) for 1 h. VO was then deacetylated by hydroxylamine (150 mM) overnight at 4 °C. Labeled antibodies were purified on a size exclusion column and eluted in PBS at a final concentration of 0.1 µM. The degree of labeling was estimated to be 6.8 dyes per antibody by absorption spectroscopy.

**Cell staining with CFl and VO ligands**. PC12 cells were washed with Fluorobrite media (Fluorobrite DMEM (Invitrogen) supplemented with 5% fetal bovine serum, 5% horse serum, 4 mM L-glutamine, 1% pyruvate, and 1% penicillin/streptomycin), and incubated with 6 µM CFl-SNAP-tag ligand (**4**) or VO-SNAP-tag ligand (**5**) in Fluorobrite media for 3 h at 37 °C. Cells were subsequently washed thoroughly, and incubated in media for an additional 2 h before imaging. SVs were labeled with

CFl-SNAP-tag ligand (**4**), VO-SNAP-tag ligand (**5**), and anti-Syt1-VO antibody. Neurons were exposed to 10 µM CFl-SNAP-tag ligand (**4**) or VO-SNAP-tag ligand (**5**) for 1 h in conditioned culture medium, washed thoroughly and then placed back in conditioned culture medium for at least 30 min before imaging. Neurons were incubated with 10 nM anti-Syt1-VO in a 37 °C incubator for 3 h in a carbonate buffer containing 105 mM NaCl, 20 mM KCl, 2.5 mM $CaCl_2$, 1 mM $MgCl_2$, 10 mM glucose, 18 mM $NaHCO_3$. Cells were then washed three times before imaging.

**Live cell imaging and analysis of PC12 cells**. Cells were imaged in buffer containing 130 mM NaCl, 2.8 mM KCl, 5 mM $CaCl_2$, 1 mM $MgCl_2$, 10 mM HEPES and 10 mM glucose. pH was adjusted to 7.4 with 1 N NaOH. The stimulation buffer contained 50 mM NaCl, 105 mM KCl, 5 mM $CaCl_2$, 1 mM $MgCl_2$, 10 mM HEPES and 1 mM $NaH2PO_4$. pH was adjusted to 7.4 with 5 M KOH. Experiments were carried out at 25 °C using TIRF microscopy[14, 23]. Cells were imaged on an inverted fluorescent microscope (IX-81, Olympus), equipped with a ×100, 1.45 NA objective (Olympus). Lasers (488 and 561 nm) (Melles Griot) were combined and passed through a LF405/488/561/635 dichroic mirror. The laser was controlled with an acousto-optic tunable filter (Andor). Emitted light was separated using a 565 DCXR dichroic mirror on the image splitter (Photometrics), and projected through 525Q/50 and 605Q/55 filters onto the chip of an EM-CCD camera. Image acquisition was done using the Andor IQ2 software. Images were acquired sequentially with alternate 488 and 561 nm excitation at 100 ms exposure. The red and green images were aligned post acquisition using projective image transformation[14, 23]. Before experiments, 100 nm yellow–green fluorescent beads (Invitrogen) were imaged in the green and red channels, and superimposed by mapping bead positions.

Image analysis was performed using Metamorph (Molecular Devices) and custom scripts on MATLAB (Mathworks). The co-ordinates of the brightest pixel in the first frame of each fusion event in the green channel was identified by eye, and time was normalized to 0 s. A circular ROI of 6 pixels (~990 nm) diameter and a square of 21 pixels (~3.5 µm) were drawn around the fusion co-ordinates. The average minimum pixel intensity in the surrounding square from five frames before fusion was subtracted from the intensity in the circular ROI, and the values were normalized to the frame before fusion in the green and red channels.

**Live cell imaging and analysis of hippocampal neurons**. The experiments with neurons were carried out in a buffer solution containing 120 mM NaCl, 5 mM KCl, 2 mM $CaCl_2$, 2 mM $MgCl_2$, 5 mM glucose, 10 mM HEPES adjusted to pH 7.4 and 270 mOsmol/l. Neurons were stimulated by electric field stimulation (platinum electrodes, 10 mm spacing, 1 ms pulses of 50 mA, and alternating polarity at 20 Hz) applied by constant current stimulus isolator (SIU-102, Warner Instruments) in the presence of 10 µM 6-cyano-7-nitroquinoxaline-2,3-dione (CNQX) and 50 µM D,L-2-amino-5-phosphonovaleric acid (AP5) to prevent recurrent activity.

Experiments were performed on an inverted microscope (IX83, Olympus) equipped with an Apochromat N oil ×100 objective (NA 1.49). Images were acquired with an electron multiplying charge coupled device camera (QuantEM:512SC; Roper Scientific) controlled by MetaVue7.1 (Roper Scientific). Samples were illuminated by a 473-nm laser (Cobolt) for green imaging, as well as by a coaligned 561-nm laser (Cobolt) for red imaging. Emitted fluorescence was detected after passing filters (Chroma Technology Corp.): 595/50 nm for pHuji, CFl and VO imaging, and 525/50 nm for SEP/GFP imaging. Simultaneous dual color imaging was achieved using a DualView beam splitter (Roper Scientific). To correct for $x/y$ distortions between the two channels, images of fluorescently labeled beads (Tetraspeck, 0.2 µm; Invitrogen) were taken before each experiment and used to align the two channels[5]. Time lapse images were acquired at 1 or 2 Hz with integration times from 50 to 150 ms.

Image analysis was performed with custom macros in Igor Pro (Wavemetrics) using an automated detection algorithm[22]. The image from the time series showing maximum response during stimulation was subjected to an "à trous" wavelet transformation. All identified masks and calculated time courses were visually inspected for correspondence to individual functional boutons. The intensity values were normalized to the 10 frames before stimulation in the green and red channels. Photobleaching in the red channels was corrected using an exponential decay fit applied on the non-responsive boutons. All data are represented as mean ± s.e.m. of the specified number of replicates in text.

**Data availability**. All primary data and analysis are available from the authors upon request. The plasmids VAMP2-pHuji, VAMP2-SNAP-tag, VGluT1-pHuji, VGluT1-SNAP-tag, VAChT-pHuji, and VAChT-SNAP-tag are available on Addgene.

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

## Acknowledgments

We thank the cell culture core facility of IINS for preparing neuronal cultures, Marie-Paule Strub (NIH) for assistance with molecular biology, and Ronak Patel and John Macklin (Janelia) for the fluorophore photobleaching experiments. This work was supported by the Agence Nationale de la Recherche (to D.P.), the ERC (to D.C.), the Intramural Research Program of the National Heart, Lung, and Blood Institute, NIH (to J.W.T.), and the Howard Hughes Medical Institute (to L.D.L.). M.M. is the recipient of a Marie Skłodowska-Curie Individual Fellowship (IF) under the Horizon 2020 Program (H2020) of the European Commission.

## Author contributions

M.M. performed and analyzed experiments on neurons. A.S. performed experiments on PC12 cells, and A.S. and J.W.T. analyzed the data. J.B.G. performed organic synthesis. T.D.G. prepared and analyzed protein conjugates. L.D.L. performed spectroscopy. M.M.,

L.D.L., J.W.T. and D.P. wrote the manuscript and all the other authors edited the manuscript.

## Additional information

**Competing interests:** L.D.L. and J.B.G. have filed patent applications whose value might be affected by this publication. The remaining authors declare no competing financial interests.

