## [Peer Review File · Nature Communications]

Reviewers' comments:

Reviewer #1 (Remarks to the Author):

I find this a timely, important and overall excellent manuscript. The technical possibility that it introduces is relevant to many groups working on synaptic function.

I only have two minor suggestions:

- the authors may want to add a table, or figure panel that compares, side by side, the performance of their new dyes with that of the red pH-sensitive protein pHuji (which they introduced in the past), and pHtomato (Li and Tsien, Nature Neuroscience, 2012).
- the authors could add a further example of synaptic analyses, adding their construct on proteins such as synaptotagmin, the glutamate transporter, or synaptophysin, as multiple groups have performed in the past with pHluorin. This would convince even further the potential user groups of the flexibility of the experiments proposed here.

Reviewer #2 (Remarks to the Author):

pH sensitive fluorescent probes, either small molecules or genetically-encoded proteins, are useful tool to investigate biochemical reactions in cells, especially in exo-/endocytosis. Martineau et al developed a set of fluorophores that are pH sensitive, amenable to live cell conjugation, and with different color spectrums. The authors went ahead and demonstrated the probes' utilities in exo-/endocytosis in a number of systems. The probe's performance is comparable to genetically-encoded green pHlurin. The work is of general interests to biologists, especially neuroscientists.

A few extra points that the author may want to address:

- 1) Photo-stability of the probes could be illustrated, e.g. comparing with a standard fluorescein or either GFP, preferably in cells.
- 2) Quantum yields and extinction coefficients can be quantified, at least at a fixed pH.
- 3) The CFI/VO-SNAP-tag ligand offers a hybrid approach to genetically label intracellular proteins in live cells. There is a concern about the non-specific labeling as the pH sensor is not fluorigenic. One could potentially use NH₄Cl to quantify the fluorescence responses as a quantitative way to assess the labeling efficiency (or non-specificity) in comparison to VAMP2-SEP.

Reviewer #3 (Remarks to the Author):

Review for Nature Communications NCOMMS-17-07115-T

In this manuscript Perrais et al. described a new technique for imaging exocytosis and endocytosis based on pH-sensitive fluorophores that they recently developed. The functions of their fluorescent probes are almost the same to so far developed pHluorins, pH-sensitive green fluorescent protein variants, which emit strong fluorescence under neutral pH but not under acidic pH condition in synapto vesicles, therefore exocytosis of vesicles can be imaged as increased fluorescence intensity. Instead of green fluorescent proteins, they utilized their own pH-sensitive and red fluorescent carbofluoresceins and tagged them to synaptic proteins via SNAP-tag technology. The imaging system looks fine, and could visualize exocytosis and endocytosis almost as pHluorins.

I could agree with the authors' claim that longer excitation wavelengths cause less phototoxicity and low autofluorescence, and make multicolor imaging possible. However, in my opinion, their probes in this manuscript are mere combinations of already developed pH-sensitive dyes and SNAP-tag technology which can be easily conceivable, and are not worth to be published in Nature Communications. Also, the same group reported novel pH-sensitive RFPs called pHuji and pHoran4 which overcame the pKa problem of so far developed pH-dependent RFPs like pHTomato, and were capable of detection of single exocytosis and endocytosis events (ref. 5). Then, what are the advantages of SNAP-VO/CFI over pHuji and pHoran4? Thorough comparison between semisynthetic and fluorescent protein-based probes should be indispensable for giving high scores for this manuscript, but there found no strict comparisons. Only I could find is the data in figure 2, but because the legends are incorrect (there are many mismatches between the descriptions and the labels), I could not properly evaluate them. Judging from the data of pKa and nH described in this manuscript and ref. 5, pHuji and pHoran4 should have almost the same capabilities to SNAP-VO/CFI, but there are no concrete discussions. Based on the above points of view, this reviewer thinks that this manuscript should be transferred to more specialized journals.

Detailed answers to reviewers' comments

Reviewer #1 (Remarks to the Author):

I find this a timely, important and overall excellent manuscript. The technical possibility that it introduces is relevant to many groups working on synaptic function.

We thank the reviewer for their very positive evaluation of our work.

I only have two minor suggestions:

- the authors may want to add a table, or figure panel that compares, side by side, the performance of their new dyes with that of the red pH-sensitive protein pHuji (which they introduced in the past), and pHtomato (Li and Tsien, Nature Neuroscience, 2012).

Thank you for this suggestion. We have now included the table shown below with the values for the pH sensitive red fluorophores we use in this work as a Supplemental Table.

Fluorophore	Emission Peak (nm)	Extinction coeff ($M^{-1}cm^{-1}$)	Quantum Yield	Source
pHuji	598	31000	0.22	Shen et al. 2014
pHoran4	561	83000	0.66	Shen et al. 2014
Carbofluorescein	567	108000	0.62	Grimm et al. 2016
Virginia Orange	581	90900	0.40	Grimm et al. 2016

- the authors could add a further example of synaptic analyses, adding their construct on proteins such as synaptotagmin, the glutamate transporter, or synaptophysin, as multiple groups have performed in the past with pHluorin. This would convince even further the potential user groups of the flexibility of the experiments proposed here.

Thank you for this suggestion. In addition to the VAMP2-SNAPtag and VAcT-SNAPtag constructs, which were described in the first version of the manuscript, we have now generated two additional probes VGLUT1-SNAPtag and VGLUT1-pHuji. These constructs are based on the vesicular glutamate transporter, which is another commonly used marker to track synaptic vesicle exocytosis. The data are presented in Supplementary Fig. 3k-m. We hope that the addition of this new probe further enhances the general utility of our method.

Reviewer #2 (Remarks to the Author):

pH sensitive fluorescent probes, either small molecules or genetically-encoded proteins, are useful tool to investigate biochemical reactions in cells, especially in exo-/endocytosis. Martineau et al developed a set of fluorophores that are pH sensitive, amenable to live cell conjugation, and with different color spectrums. The authors went ahead and demonstrated the probes' utilities in exo-/endocytosis in a number of systems. The probe's performance is comparable to genetically-encoded green pHluorin. The work is of general interests to biologists, especially neuroscientists.

We thank the reviewer for their very positive evaluation of our work.

A few extra points that the author may want to address:

1) Photo-stability of the probes could be illustrated, e.g. comparing with a standard fluorescein or either GFP, preferably in cells.

We have performed in vitro bleaching curves for fluorescein, CFI and VO. We now show that the latter is ~ 3 times more photostable than the other two dyes (Supplementary Fig. 1). We felt that the in vitro bleaching experiments would provide the most straight-forward comparison between the three dyes in a controlled environment without the complications of a cellular environment.

2) Quantum yields and extinction coefficients can be quantified, at least at a fixed pH.

Thank you for this suggestion. The quantum yields and extinction coefficients for these dyes were reported in previous publications and we now summarize these parameters as a Supplemental Table (please refer to Reviewer#1's comments). We hope this improves the manuscript.

3) The CFI/VO-SNAP-tag ligand offers a hybrid approach to genetically label intracellular proteins in live cells. There is a concern about the non-specific labeling as the pH sensor is not fluorescent. One could potentially use NH₄Cl to quantify the fluorescence responses as a quantitative way to assess the labeling efficiency (or non-specificity) in comparison to VAMP2-SEP.

We agree that non-specific and background labeling with exogenous dyes can be problematic. Thus, we now performed calibrations with application of NH₄ to normalize the fluorescence changes. The three red probes (VAMP2-pHuji/SNAP-CFI/SNAP-VO) report the same fraction of fluorescence (i.e. exocytosed vesicles) as VAMP2-SEP after electrical stimulation. These new data, shown in Supplementary Fig. 3a-f, indicate that potential non-specific binding of the SNAP ligands in this system is minimal.

Reviewer #3 (Remarks to the Author):

Review for Nature Communications NCOMMS-17-07115-T

In this manuscript Perrais et al. described a new technique for imaging exocytosis and endocytosis based on pH-sensitive fluorophores that they recently developed. The functions of their fluorescent probes are almost the same to so far developed pHluorins, pH-sensitive green fluorescent protein variants, which emit strong fluorescence under neutral pH but not under acidic pH condition in synapto vesicles, therefore exocytosis of vesicles can be imaged as increased fluorescence intensity. Instead of green fluorescent proteins, they utilized their own pH-sensitive and red fluorescent carbofluoresceins and tagged them to synaptic proteins via SNAP-tag technology. The imaging system looks fine, and could visualize exocytosis and endocytosis almost as pHluorins.

I could agree with the authors' claim that longer excitation wavelengths cause less phototoxicity and low autofluorescence, and make multicolor imaging possible. However, in my opinion, their probes in this manuscript are mere combinations of already developed pH-sensitive dyes and SNAP-tag technology which can be easily conceivable, and are not worth to be published in Nature

Communications. Also, the same group reported novel pH-sensitive RFPs called pHuji and pHoran4 which overcame the pKa problem of so far developed pH-dependent RFPs like pHtomato, and were capable of detection of single exocytosis and endocytosis events (ref. 5). Then, what are the advantages of SNAP-VO/CFI over pHuji and pHoran4? Thorough comparison between semisynthetic and fluorescent protein-based probes should be indispensable for giving high scores for this manuscript, but there found no strict comparisons. Only I could find is the data in figure 2, but because the legends are incorrect (there are many mismatches between the descriptions and the labels), I could not properly evaluate them. Judging from the data of pKa and nH described in this manuscript and ref. 5, pHuji and pHoran4 should have almost the same capabilities to SNAP-VO/CFI, but there are no concrete discussions.

Based on the above points of view, this reviewer thinks that this manuscript should be transferred to more specialized journals.

We thank the reviewer for these comments. We apologize for the mislabeling of Figure 2 panels which we have now corrected.

There are several distinct advantages offered by the small organic fluorophores CFI and VO over the existing genetically-encoded pH sensitive red fluorescent proteins, namely pHuji.

First, CFI and VO have improved sensitivity to detect changes in pH from 5.5 (the intraluminal pH of secretory vesicles and recycling endosomes) to 7.4 (extracellular pH). Like SEP/pHluorin, they are virtually non-fluorescent at low pH and become brightly fluorescent at neutral pH. Even though pHuji and other RFPs such as pHtomato are able to report exocytosis (this study and Li & Tsien, 2012), we show here that single exocytic events in PC12 cells are harder to detect with VAcHT-pHuji than with VAcHT-pHluorin or VAcHT-SNAP-CFI/VO (Figure 2). Therefore, for high resolution single vesicle exocytosis studies, these new fluorophores offer a great improvement when red fluorescence is needed for multi-color, low auto-fluorescence background, or deep tissue imaging applications. Second, small organic fluorophores can be used to label high affinity ligands and antibodies. This raises the option to label endogenous proteins without overexpression of a reporter protein (pHluorin or pHuji). We demonstrate this possibility with antibodies directed against the extracellular epitope of synaptotagmin 1 labelled with VO (anti-Syt1-VO). After incubating neurons in depolarizing buffer and rinsing, we can detect exocytosis in transfected and untransfected neurons (Figure 3e-f). This is a new feature of these fluorophores that opens new possibilities to study the trafficking of endogenous receptors. These studies could be extended to nanobody-labeling technology to further enhance the utility of these new probes.

Finally, the ability to titrate the amount of fluorophore to a preparation will allow for future developments in single molecule imaging or super-resolution imaging for nanoscale tracking of individual proteins released from single vesicles in living cells.

We believe that these new semi-synthetic tools are a significant advance to the available methods to study exocytosis and endocytosis in living cells. The increase in brightness and dynamic range, spectral range, flexibility of labeling, reduced size, and sequential labeling possibilities, are all powerful advantages that will be useful to the general neuroscience and cell biology community. Thus, we believe our work will be of interest to the wide readership of Nature Communications.

Reviewers' Comments:

Reviewer #1 (Remarks to the Author):

The authors have replied to all of my comments, and I am happy to suggest that the manuscript be published in the current form.

Reviewer #2 (Remarks to the Author):

The author generally address my concerns.

One minor thing I wish authors could present is the photostability of these probes inside the cells. In addition, the author present the in vitro photobleaching curves without error bars or statistics.

The author should also deposit the relevant plasmids used in this MS (I understand, probably not the consumable small chemical probes) into sharing centers, e.g. addgene for distribution.

Reviewer #3 (Remarks to the Author):

I looked over the revise manuscript, and found the authors revised the manuscript by partly addressing the issues I raised before. However, I am disappointed that they did not provide the revision about the novelty of the organic probes, which should be quite critical to accept in Nature sister journals.

Although I am not fully satisfied due to the above-mentioned issue, other two reviewers had more positive opinions on this manuscript which I can partly agree, I agree here to accept this manuscript in Nature Communications.

Answers to reviewer's comments

Reviewer #1 (Remarks to the Author):

The authors have replied to all of my comments, and I am happy to suggest that the manuscript be published in the current form.

Thank you!

Reviewer #2 (Remarks to the Author):

The author generally address my concerns.

Thank you for your positive evaluation.

One minor thing I wish authors could present is the photostability of these probes inside the cells. In addition, the author present the in vitro photobleaching curves without error bars or statistics.

We have added the mean \pm SD of three measurements for the three fluorophores to the in vitro bleaching curves and τ_{bleach} values in Supplementary Figure 1a. We did not provide a systematic comparison of bleaching rates for the data in cells because the laser intensities and camera settings were not kept the same throughout the experimental sessions. Therefore a direct comparison is not possible.

The author should also deposit the relevant plasmids used in this MS (I understand, probably not the consumable small chemical probes) into sharing centers, e.g. addgene for distribution.

We are in the process of submitting the plasmids coding for the following constructs to Addgene: VAMP2-pHuji; VAMP2-SNAP-tag; VGlut1-pHuji; VGlut1-SNAP-tag; VChT-pHuji; VChT-SNAP-tag. They should be available soon.

Reviewer #3 (Remarks to the Author):

I looked over the revise manuscript, and found the authors revised the manuscript by partly addressing the issues I raised before. However, I am disappointed that they did not provide the revision about the novelty of the organic probes, which should be quite critical to accept in Nature sister journals.

Although I am not fully satisfied due to the above-mentioned issue, other two reviewers had more positive opinions on this manuscript which I can partly agree, I agree here to accept this manuscript in Nature Communications.

We are sorry to read that our arguments did not fully convince the reviewer on the new possibilities brought by the new pH sensors. We did add a sentence explaining that although CFI and VO were used as scaffolds for fluorogenic substrates, this is the first report of using them as biomolecule labels. Nevertheless, we thank her/him for accepting the publication of our manuscript in Nature Communications.